# Oral Squamous Cell Carcinoma-Associated Thrombosis: What Evidence?

**DOI:** 10.3390/cancers14225616

**Published:** 2022-11-16

**Authors:** Leonardo Di Gennaro, Raimondo De Cristofaro, Antonietta Ferretti, Maria Basso, Claudia Riccio, Massimo Cordaro, Carlo Lajolo

**Affiliations:** 1Hemorrhagic and Thrombotic Diseases Center, Department of Translational Medicine and Surgery, Fondazione Policlinico Universitario “Agostino Gemelli” IRCCS, 00168 Roma, Italy; 2Chimica, Biochimica e Biologia Molecolare Clinica, Fondazione Policlinico Universitario “Agostino Gemelli” IRCCS, 00168 Roma, Italy; 3Head and Neck Department, Institute of Dentistry and Maxillofacial Surgery, Fondazione Policlinico Universitario “Agostino Gemelli” IRCCS, 00168 Roma, Italy

**Keywords:** oral squamous cell carcinoma, oral cancer, venous thromboembolism, cancer-associated thrombosis

## Abstract

**Simple Summary:**

The relevance of cancer-associated thrombosis has been increasingly recognised, both by physicians who treat patients with venous thromboembolism and oncologists. Active cancer accounts for 20% of the overall incidence of venous thromboembolism. While oral squamous cell carcinoma is the most common form of oral cancer, its relationship with venous thromboembolism rarely has been investigated. In this review, we provide an overview of this topic, with the aim of raising awareness on its relevance for patient care.

**Abstract:**

Venous thromboembolism (VTE) disease is the second leading cause of mortality in cancer patients. In the general population, the annual incidence of a thromboembolic event is about 117 cases per 100,000 persons, but cancer increases this risk about fourfold, while in patients receiving chemotherapy and surgical treatment, it is about sevenfold. Oral squamous cell carcinoma (OSCC) is the most common form of oral cancer and represents a multistep process in which environmental factors and genetic alterations are implicated. Thrombotic risk is considered empirically low in OSCC patients, although few data are available. Having limited information available may result in poor awareness of VTE prevention in OSCC, risking jeopardising the oncologic treatment and increasing the morbidity and mortality among these patients. In this paper, the topic of OSCC-associated thrombosis will be discussed.

## 1. Introduction

It has been estimated that patients with cancer have an approximately fourfold higher risk of venous thromboembolism (VTE) [1]. VTE is the second leading cause of mortality in cancer patients, after the malignant disease itself [2]. Therefore, VTE treatment and prophylaxis are essential for the global management of patients with cancer.

Several biological mechanisms on a hypercoagulable state induced by malignant cells have been described, bringing together different neoplasms in different sites [3,4].

Oral squamous cell carcinoma (OSCC) represents more than 90% of oral cavity cancers and is the most frequent localization of head and neck cancer [5]. For this reason, in this paper, OSCC and the term oral cancer will be considered synonyms, for convenience. OSCC is the seventh most common human malignancy, and its incidence is increasing worldwide, especially among young people due to lifestyle factors such as smoking and alcohol abuse [5,6]. OSCC may also result from any chronic irritation, such as dental caries, overuse of mouthwash, chewing tobacco, or the use of betel quid. Human papillomavirus has also been identified as an independent risk factor [7].

It is imperative to diagnose OSCC early in order to achieve the best possible treatment and a favourable prognosis. The established routine check-up includes medical history, extra-oral and intra-oral examination, and, in case of clinical findings, a biopsy of the potentially malignant lesion.

Current treatment strategies for OSCC mainly include surgery and radiation therapy, and in some cases adjuvant chemotherapy before a surgical procedure [5,6,7,8]. Moreover, routine surgical reconstruction is key in order to reduce postoperative oral disabilities. Speech and swallowing therapy may be required after significant resections [8].

Chemotherapy is not used routinely as primary therapy but is recommended as adjuvant therapy along with radiation in patients with advanced disease [8,9].

Indeed, the risk of thrombosis with OSCC is not known. It is generally accepted that OSCC has a low risk of VTE, despite most mechanisms described in cancer-associated thrombosis being similar [10]. As demonstrated for other cancers, radio and chemotherapy could also increase the risk of VTE in these patients.

Our article aims to report the available findings on this topic.

## 2. Pathogenesis of Oral Cancer-Associated Thrombosis

The mechanisms underlying the occurrence of VTE in cancer can be explained by the so-called Virchow triad, which includes alterations in haemostasis, the vessel wall, and blood stasis. An important role in the pathogenesis of the hypercoagulable state associated with neoplasms is attributed to the prothrombotic properties of the cancer cells themselves [10,11]. The state of hypercoagulability may also be characterized by abnormalities in one or more laboratory tests of haemostasis, demonstrating varying degrees of subclinical activation of coagulation in association with cancer growth. These cells can activate the coagulation cascade by a direct mechanism through the production of procoagulant substances such as tissue factor (TF), which is constitutively expressed by these cells and forms a complex with coagulative factor VII to activate the coagulation cascade.

More specifically for oral cancer, TF can be considered the best characterized tumour-derived procoagulant protein. TF, also called factor III or thromboplastin, is a transmembrane protein that can initiate the clotting process by thrombin formation from the zymogen prothrombin, activating platelets [12]. With urokinase-type plasminogen activator receptor and epidermal growth factor receptor, TF can be also considered a prognostic biomarker in OSCC and could potentially be attractive targets for molecular imaging and therapy in OSCC due to its high positive expression rates and tumour-specific expression patterns [13,14].

Another important mechanism inducing coagulation was described by Adesanya and colleagues, who showed that microparticles (MPs) from OSCC cell lines in vitro could induce TF expression by endothelial cells and stimulate platelet aggregation [15]. MPs have a well-known role in thrombus formation in vivo in cancer patients and high levels of circulating MPs have been reported in patients with OSCC [16,17,18].

MPs are also the likely carriers of fibrin in the peritumoural sites of OSCC, such as in neck vessels.

As in other different neoplasms, an inflammatory response typical of OSCC may trigger thrombosis [19,20,21,22]. A high level of thromboxane A2, von Willebrand factor, and prostacyclin, as well as decreased thrombomodulin plasma levels have been reported in OSCC patients [23,24]. Moreover, a low thrombomodulin level seems to be associated with a poor prognosis [25].

Unfortunately, these mechanisms, which can only partly explain the pathogenesis of OSCC-associated thrombosis, are not supported by laboratory tests. Coagulation tests cannot help the clinician in estimating the thrombotic risk of patients with OSCC, although an increased blood d-dimer may prompt investigations to exclude thrombosis. However, in the presence of an elevated d-dimer value, Doppler echocardiography may be used to examine neck vessels and exclude, in particular, jugular vein thrombosis [6].

Different assays, such as the thrombin generation assay, could be used in future studies to analyse the thrombin generation potential in plasma of OSCC patients and contribute to assess their thrombotic risk.

## 3. OSCC and Thrombosis: A Paradoxical Relationship?

All the mechanisms described above, together with other general factors, such as the compression stasis of the tumour mass, infections, and bed rest, may contribute to the hypercoagulable state of OSCC patients. However, a paradoxical situation has been described for OSCC patients because, unlike other types of cancer, they seem to present a low incidence of thrombosis [1,2,3].

In a recent study, a potential mechanism that may explain this paradox was explained using a mouse model of tongue squamous cell carcinoma [26]. Surprisingly, platelet accumulation and fibrin generation in OSCC were highly inhibited in tumour-bearing mice versus healthy controls, suggesting a possible reduction in blood clot formation. The authors explained this mechanism with a delta-storage pool deficiency, probably due to a qualitative platelet defect. However, the mechanism involved in the acquired delta-storage pool deficiency remains unclear and has not been confirmed in human models or elsewhere [27].

The most well-known independent risk factors for cancer-associated thrombosis are age, immobility due to hospitalization, previous history of thrombosis, indwelling catheters, surgery, and chemotherapeutic agents. Some of these risk factors are common among OSCC patients, but, unfortunately, few data are available, and many confounding factors are present in those studies, which contribute to the unexpected findings regarding this topic [28,29]. Recently, a literature review confirmed OSCC as having one of the lowest incidences of thrombosis among cancers; however, even in this review, the authors revealed that there still is a lack of clarity on the association between OSCC and thrombosis [10].

In our clinical experience, jugular vein thrombosis is the most frequent in oral cancer patients (7 cases out of 42 patients; data not published). This venous mass may not be a fibrin clot, but rather a mass of tumour cells, developed from tumour mass invasion and penetration into the lumen of local veins and can be the first step of systemic dissemination of tumour cells. In fact, in all our patients, a CT-PET scan confirmed local lymph node progression. As reported elsewhere, the histopathological presence of lymphovascular invasion has a poorer prognostic outcome [7,8,9,10]. Moreover, in our experience, surgical failure was more common in patients with thrombosis of the external and/or internal jugular vein, especially after microvascular reconstruction.

Furthermore, surgery of oral cancer often involves long procedures, especially in cases of simultaneous reconstruction, causing prolonged hospitalization and bed rest. In a retrospective study of patients with head and neck cancer, the incidence of VTE in patients who underwent reconstruction after cancer resection ranged from 1.4 to 5.8%, and the main predictors for VTE were old age, prior thrombosis, obesity, and blood transfusion. However, this incidence was lower among patients with oral cancer [30].

The incidence of postoperative VTE in these patients was studied only in one prospective study with a high (26%) but not statistically significant number of thromboses [31]. Of note, this study tested the association of the Caprini Risk Score (CRS) with VTE (Figure 1). Univariate logistic regression analysis showed that female gender and a high CRS were meaningfully associated with a higher frequency of VTE occurrence, but a multivariate logistic regression analysis showed that only a high CRS was statistically relevant to VTE occurrence. Therefore, we agree with the authors in suggesting lower-limb Doppler ultrasound for patients undergoing oral cancer surgery with a high risk according to the CRS.

Unlike thrombosis of the lower limbs or thrombosis of the neck vessels, a literature search on the association between OSCC and pulmonary embolism did not return results. Certainly, as with other cancers, in OSCC, an asymptomatic PE incidentally detected during restaging is a challenge.

An important remark must be made regarding the medical treatment of these patients. Chemotherapeutic agents have been associated with an increased risk of arterial and venous thrombosis [33]. Indeed, in a predictive model, chemotherapy is as critical as the site of the cancer and other clinical and laboratory complications (Table 1) [34].

Many types of chemotherapy drugs can be associated with an increased risk of thrombotic events, including patients with oral cancer (Table 2) [35,36,37].

Especially among these patients, cisplatin seems to increase the TF activity and high blood level of circulating MPs [38,39]. An association between 5-fluoruracil and higher thrombotic risk also has been reported [40,41]. In another study, carboplatin plus oral tegafur and cetuximab resulted in a safe, well-tolerated first-line therapy for recurrent or metastatic OSCC, with a very low incidence of thrombosis [42].

In conclusion, these data seem to demonstrate that the thrombosis risk associated with oral cancer is low. Nevertheless, we believe that there are so many contradictions and so many “shadow areas” that only focused studies can resolve. While waiting for prospective clinical studies, in our opinion, the same attention to thrombosis risk should apply to these patients, as we do with patients with other types of cancer. Therefore, general recommendations about VTE prevention and management in oncologic patients have to be considered according evidence-based guidelines [43,44]. About this topic, guidelines for surgical head and neck cancer patients are also available [45,46].

In accordance with the above, we suggest a potential strategy regarding how to investigate and uncover the risk of VTE in OSCC patients (Figure 2).

## 4. Remarks on Prevention and Treatment of OSCC-Associated Thrombosis

There are no published studies on the prevention and treatment of VTE in OSCC. Therefore, what we report is derived from the management of the cancer–thrombosis association. Generally, therapeutic anticoagulation in patients with cancer-associated VTE requires carefully balancing the risks and benefits. The management of prevention and treatment of VTE in patients with active cancer is challenged by a high risk of both recurrent VTE and bleeding events, and the choice of anticoagulant is often a problem, especially in the case of prophylaxis. To date, low-molecular-weight heparins (LMWH) or fondaparinux represent the standard of care for the prevention and treatment of cancer-associated VTE, although the safety and efficacy of direct oral anticoagulants (DOAC) is increasingly emerging [47,48,49,50].

In fact, the American Society of Clinical Oncology guidelines recommend prophylaxis or therapy with LMWH and DOAC (rivaroxaban, apixaban) for high-risk patients with cancer and in case of VTE (see the Khorana score in Table 1), when no significant bleeding risk factors or drug–drug interactions are evident. Based on recent clinical studies, anticoagulation with either apixaban, edoxaban, or rivaroxaban has been incorporated in the updated guidelines for the management and treatment of cancer-associated VTE as alternative to LMWH [43,51,52]. Unfortunately, all of these indications can be applied to oral cancer despite few patients being enrolled. For example, in the Caravaggio study, only 14 out of 576 patients (2.4%) were included with this type of cancer [50].

According to these guidelines, most hospitalized cancer patients should be subjected to thromboprophylaxis. This recommendation does not apply to ambulatory cancer patients.

In cases of surgery, which is a frequent option for oral cancer, we agree with thromboprophylaxis being started 12 h before surgery and continued for at least 7–10 days after.

Indeed, thrombosis can be a complication in oral cancer surgery and may be associated with flap failure. Therefore, even though no consensus in the literature has been found on how thrombosis could best be prevented, we suggest thromboprophylaxis to prevent thrombosis in the anastomosis, according to the general guidelines.

In our opinion, outside of surgery, a medical thromboprophylaxis is correct only in the absence of a high bleeding risk and in the presence of at least three important VTE risk factors (see Figure 2). A mechanical prophylaxis for VTE for all these patients can be suggested, but which typically have poor efficacy.

Treatment of VTE in OSCC patients should regularly be clinically monitored to optimise therapy compliance. Patients should be treated for 3 months at least, and continue anticoagulation for secondary prevention of VTE, if the cancer is not in complete remission and/or anti-cancer therapies are ongoing.

In incidental PE, due to high morbidity and mortality, anticoagulation always should be always initiated in OSCC patients, even though segmental or more distal subsegmental branches are involved. The advantage of DOAC lies in its oral administration and easy, standardised dosing, so enhancing patient compliance. Nevertheless, LMWH could be the first option for VTE associated with OSCC also because of possible difficulties in swallowing for these patients. Moreover, DOAC safety and efficacy can further be influenced by drug–drug interactions. DOAC are substrates of the P-glycoprotein (P-gp) and are in part metabolised through the CYP3A4 enzyme. Therefore, in this scenario, anticoagulant plasma levels can be altered by some anti-cancer drugs that interfere with CYP3A4 or P-gp [51,52].

## 5. Conclusions

The pathogenesis of thrombosis associated with oral cancer is complex and involves both clinical and biological factors. Moreover, VTE is an important cause of morbidity and mortality in patients with malignancy, and this also applies to OSCC. Even though VTE is considered less frequent in OSCC than in other cancers, to date, no strong factual evidence exists of a poor association between OSCC and VTE. On the contrary, OSCC shows similar features to tumours with a high thrombotic risk.

In this chapter, we have identified important clinical questions linked to OSCC and provide recommendations to clinicians based on analysis of the limited data available. A weak aspect of this article is certainly the lack of publications, many important issues remain to be addressed regarding this topic; in particular, how best to enhance appropriate utilization of outpatient thromboprophylaxis in OSCC and how to stratify the thrombotic risk in oral cancer. We believe that in the meantime, stratifying the thrombosis risk may be useful using validated scores, and have suggested performing targeted investigations in the presence of specific risk factors. In this scenario, the use of biological tests directly based on the molecular features of coagulation might be useful in determining the real thrombotic risk associated with oral cancer.

Considering the scientific interest in this area that has emerged in the last years due to increasing incidences of OSCC, we are sure that the scientific community can continue to identify ways to enhance patient-centred care by commencing larger clinical trials.

## Figures and Tables

**Figure 1 cancers-14-05616-f001:**
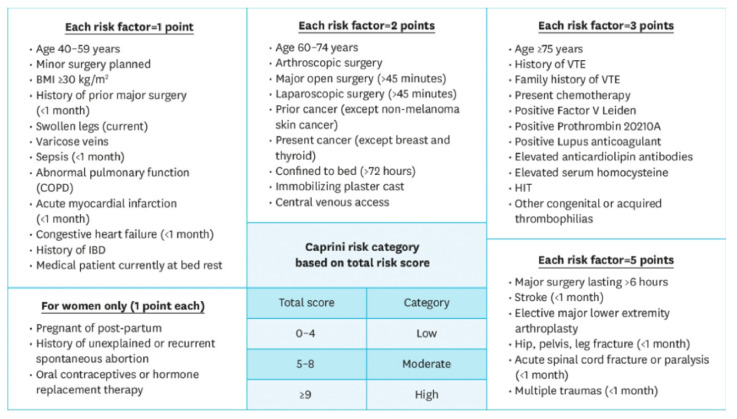
Caprini Risk Score (RCS). The Caprini Risk Score was developed to provide an adequate and individualized VTE prophylaxis risk assessment model. The score assigns points based on 20 risk factors taken from a patient’s past medical history as well as their current health, in which a higher score classifies a patient at a higher risk. Previous studies in other surgical specialties, such as otolaryngology, bariatrics, thoracic surgery, and plastic reconstructive surgery, have demonstrated the predictive efficacy of the Caprini Risk Score [32].

**Figure 2 cancers-14-05616-f002:**
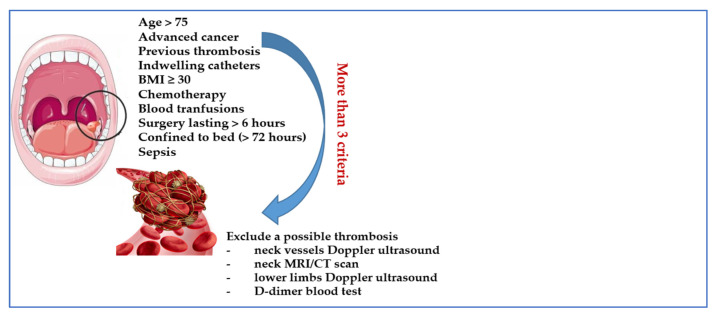
Suggested criteria for investigating asymptomatic thrombosis in OSCC.

**Table 1 cancers-14-05616-t001:** The Khorana score for estimating venous thromboembolism risk in patients with cancer.

Risk Factor	Points
Site of primary tumour	
Very high risk (stomach, pancreas)	2
High risk (lung, lymphoma, gynaecologic, bladder, testicular)	1
All other sites	0
Pre-chemotherapy platelet count ≥ 350,000/microL	1
Haemoglobin level < 10 g/dL or use of ESAs	1
Pre-chemotherapy WBC > 11,000/microL	1
BMI ≥ 35 kg/m^2^	1

Khorana score points: 0 (low); 1–2 (intermediate); ≥3 (high). ESAs: erythropoiesis-stimulating agents; WBC: white blood cell; BMI: body mass index; VTE: venous thromboembolism.

**Table 2 cancers-14-05616-t002:** Anticancer agents and thrombosis.

	Agent	VTE	ATE
Platinum-based agents	Cisplatin	++	−
	Carboplatin	++	−
	Oxaliplatin	+	−
Anthracyclines	Doxorubicin	+	NR
	Daunorubicin	NR	NR
	Epirubicin	NR	NR
Pyrimidine antagonists	5-fluorouracil	−	−
	Gemcitabine	−	−
	L-asparaginase	NR	NR
	Tamoxifen	+	+
Immunomodulatory agents	Thalidomide	++	NR
	Lenalidomide	++	NR
	Pomalidomide	+	NR
Anti-EGFR antibodies	Cetuximab	+	−
	Panitumumab	+	−
	Necitumumab	+	−
VEGF-targeted molecules	Bevacizumab	−	++
	Aflibercept	−	NR
VEGFR RTKI	Sunitinib	−	++
	Sorafenib	−	++
	Axitinib	−	++
	Pazopanib	−	+
	Vandetanib	−	+
	Lenvatinib	NR	NR
	Cabozantinib	NR	NR
BCR-ABL RTKI	Imatinib	−	−
	Dasatinib	++	++
	Nilotinib	++	++
	Ponatinib	++	++
	Bosutinib	NR	NR
CDK inhibitors	Palbociclib	+	NR
	Abemaciclib	++	NR
	Ribociclib	+	NR

+ associated; − not associated. ATE, arterial thromboembolism; BCR-ABL, breakpoint cluster region protein-tyrosine kinase protein ABL1; CDK, cyclin-dependent kinase; EGFR, epidermal growth factor receptor; NR, not reported; RTKI, receptor tyrosine kinase inhibitor; VEGF, vascular endothelial growth factor; VEGFR, VEGF receptor; VTE, venous thromboembolism.

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
