# Peer review of "Oral Squamous Cell Carcinoma-Associated Thrombosis: What Evidence?"

_cancers, 2022, doi:10.3390/cancers14225616_

Round 1
Reviewer 1 Report
Gennaro et al aimed at discussing the topic of oral squamous cell carcinoma-associated thrombosis. In this manuscript, they summarized current literatures regarding the association of OSCC with venous thromboembolism (VTE). However, the related topic has been reported by Haen et al (Biomedicines, 2021), who comprehensively summarize the incidence of head and neck cancer following VTE. In this regard, the manuscript looks like no novelty. In addition, most of the description are about the basis of VTE, the risk assessment, and the related anti-cancer drugs in a wide variety of cancers. The author should look for any kinds of associated literatures (including case reports) related to OSCC with thrombosis and list them. The author should also include potential strategy regarding how to investigate and uncover the risk of VTE in OSCC with an original Figure(s).
Author Response
Di Gennaro et al aimed at discussing the topic of oral squamous cell carcinoma-associated thrombosis. In this manuscript, they summarized current literatures regarding the association of OSCC with venous thromboembolism (VTE).
- However, the related topic has been reported by Haen et al (Biomedicines, 2021), who comprehensively summarize the incidence of head and neck cancer following VTE. In this regard, the manuscript looks like no novelty…
We would like to thank the reviewer for his critical comments, which enabled the paper to be improved.
We agree with the reviewer about the importance of Haen's article published in Biomedicines. However, in that paper, which we often refer in our article, the authors specifically discuss an experimental animal study showing a possible association between reduced platelet activity and a low risk of systemic thrombosis in tongue cancer. We think of offering to Cancers readers further “food for thought” on the issue regarding association between thrombosis and oral cancer generally, with the hope of promoting further clinical studies. We regret that our paper offers no novelty on this topic. Therefore, according to reviewer’s suggestions, we have made an effort making the manuscript more appealing with original data based on our clinical experience.
- In addition, most of the description are about the basis of VTE, the risk assessment, and the related anti-cancer drugs in a wide variety of cancers. The author should look for any kinds of associated literatures (including case reports) related to OSCC with thrombosis and list them. The author should also include potential strategy regarding how to investigate and uncover the risk of VTE in OSCC with an original Figure(s).
Lastly, we have taken up the reviewer's comments and his critical remarks. We have tried to respond to his requests, especially cutting on general aspects about cancer-associated thrombosis and by improving research in the literature on the association OSCC – thrombosis (unfortunately not so much!). Furhermore, according reviewer’s suggestion, we have emphasized the role of stratification scores as strategy to investigate and uncover the risk of VTE in OSCC and proposed a decision-making algorithm for the surgeon and clinician with an original Figure (2).
Reviewer 2 Report
The Authors present a review on the association between oral squamous cell carcinoma (OSCC) (the most common type of oral cancer) and venous thromboembolism (VTE). The main argument for this publication holds: indeed, while much is known on the association of cancer with VTE in general and of specific types of cancer (for instance, solid vs. hematogenous cancers) with VTE in particular, little has been written on the specific topic of OSCC and VTE. An overview of what little information is available would be clinically useful. In addition, the Authors present some interesting information on the biological characteristics of SCC and their potential association with thrombosis that may have implications for prevention and for future research. However, the article in its present form is not suitable for publication and should be revised before it can be considered for Cancers. The main problems are the organization, formatting, and presentation of the information. While the English is in general adequate, the article should be proofread because of some redundancies and style issues. In addition, the clinical relevance of the topic and of available evidence should be emphasized much more strongly. I present the major issue in detail below.
MAJOR ISSUES
1) While the written English is generally adequate, with no major grammar issues, the style should be considerably improved. In particular, several paragraphs and sentences display redundancy: removing unnecessary sentences or phrases would greatly increase readability, as required by scientific style. I picked a couple of examples, but these examples are non-exhaustive and addressing the examples only would not solve the problem.
> “No less important in the pathogenesis of cancer-associated thrombosis is the role of chemotherapy. The role of chemotherapy in the increase of thrombotic events in cancer patients is evident in recent decades. In fact, cancer patients have an estimated almost 7-fold increased risk of developing chemotherapy-associated thrombosis [32]. Therefore, a predictive model has been developed in which the role of chemotherapy is as critical as the site of cancer and other clinical and laboratory complications (Table 1) [33].” The three initial clauses simply repeat three times the introductory point that chemotherapy increases the risk of increasing thrombosis. That’s it. Two of the three could be removed and the paragraph would be more concise and work perfectly: “Cancer patients have an estimated almost 7-fold increased risk of developing chemotherapy-associated thrombosis [32]. Indeed, in a predictive model chemotherapy is as critical as the site of cancer and other clinical and laboratory complications (Table 1) [33].” (also, why “the role of chemotherapy” and not just “chemotherapy”, which would be more concise?).
> “OSCC and thrombosis: a paradoxical situation or not?” The “or not” is pleonastic. Think about it: it can be added to any question without adding any information (should one write “Does anticoagulation prevent thrombi?” or “Does anticoagulation prevent thrombi OR NOT?”; should one write “Does this therapy work?” or “Does this therapy work or not?”, and so on). Removing it leaves the question in the standard form: “OSCC and thrombosis: a paradoxical situation” or, much better and specifically, “OSCC and thrombosis: a paradoxical relationship?”
> “physiopathogenesis”: why not just “pathogenesis”?
> “Moreover, other factors related to cancer management, such as surgery, chemotherapy, radiotherapy, hormonal therapy, hospitalization, and indwelling venous catheters, may further increase the risk of VTE [4].” Should it not simply be “Cancer management strategies” rather than “Other factors related to cancer management”?
> “there are no studies that have largely investigated this association as in other cancers.”: “largely” is inappropriate here and can be removed (maybe the Authors meant “in detail”?).
> “The recommended treatment after VTE diagnosis should continue for three to six months”: one either writes “recommended” or “should”. Therefore, either “The recommended treatment after VTE diagnosis is three to six months” or “The treatment after VTE diagnosis should continue for three to six months”.
Therefore, before anything else, I would ask the Authors to reduce by 25% the total length of the manuscript by carefully going through it and removing unnecessary words, phrases, and sentences.
2) Clarity is further affected by at times inadequate style and formatting. Again, some instances that are non-exhaustive:
> The paragraph beginning with “Notwithstanding” is not entirely clear. “Notwithstanding risk factors overlapping with other cancers, it is very difficult to determine the risk of thrombosis for OSCC.” What do the Authors mean? Maybe, that despite the fact that the risk factors for thrombosis in OSCC are similar to those for thrombosis in other cancers (but then the second clause does not make sense)? Or “Despite the fact that the risk factors for OSCC are similar to those of other cancers, it is difficult to determine the risk for thrombosis in patients with OSCC” (but this also does not make much sense, because the fact that the risk factors for OSCC are similar to those for other cancers does not imply that it should then be easy to deduce the risk factors for VTE in cancer, which is another thing altogether)? One of the next sentences is also not quite clear: “However, the authors precisely pointed out also an unexplained paradox because biologically oral cancer is associated with most mechanisms found in cancers associated thrombosis.” Do the Authors mean “However, the authors precisely pointed out that this low risk is unexpected, because the biological mechanisms leading to oral cancer overlap considerably to those leading to cancer associated thrombosis” (I do not know whether this is the case – this is just a possible interpretation!). Please check carefully and rephrase the whole period as clearly as possible.
> The use of new lines is questionable. Each paragraph should discuss a topic or a previous study, and a new line should signal that the discussion moves to a new topic or a new study. Why does the paragraph “Microparticles (MP) are small membrane vesicles derived ...” begin with a new line, if the sentence is a clarification of the notion introduced in the previous sentence on the study by Adesanya et al.? And why is the acronym MP introduced twice, in the previous sentence and this one? Another example occurs later: the four paragraphs from “Specifically, the main thrombotic risk factors...” to “with a high risk according to Caprini score” all refer to the same study. A sentence starting with “Finally, the authors reported that...” as the FIRST sentence of a paragraph; it should be the last sentence of a paragraph, then a new line comes to move on to the next topic (chemotherapy).
> There is a proliferation of “either” in the sentence “The thrombotic event with vessel occlusion can occur either by platelet activation resulting in hyperaggregability caused either by direct contact with cancer cells or by the activation of endothelial cells themselves by MPs of cancer cells.” I have the impression that the second “either” is not necessary. Difficult to understand; one would expect, after the first an “or” after the first “either”, but then a second “either” appears. Please correct.
> there are several typos: “compression stasis by tumor masse,” (mass), “Adapted form Haen P et al” (should be: “from”). “rest bed”: it should be “bed rest”.
> “Actually...” Please remove (colloquial, not academic English).
> Word order is not always correct: for instance, “It is imperative to early diagnose OSCC”: better as “It is imperative to diagnose OSCC early”; “”including also following reconstruction” (it should probably be either “including” or “also”); “in patients with OSCC surgical candidates CRS is another score that clinicians should consider” (do the Authors mean “in patients with OSCC who are eligible for surgery clinicians should also consider the CRS scoring system”?).
> “An important mechanism inducing coagulation and involving TF was described by Adesanya et al. In this paper they showed that microparticles (MP) from OSCC cell lines in vitro could induce TF expression by endothelial cells also stimulating platelet aggregation [15].” It is not possible to write “In this paper” at a location in the sentence in which no paper has been cited (only a group of Authors). If anything, one could write “An important mechanism inducing coagulation and involving TF was described in a study by Adesanya et al., which showed that (...)”
> Commas are not always used appropriately; for instance, in “Cancer cells can activate the coagulation system, through indirect mechanisms, inducing the expression of a procoagulant phenotype.” Should it not rather be “Cancer cells can activate the coagulation system through indirect mechanisms, by inducing the expression of a procoagulant phenotype”. The first comma should not be there.
Therefore, my second suggestion is to have a professional proofreader or a native English speaker go through the article to solve these problems. They can also help with step 1 above (the reduction of 25% of the length of the article in its present form).
3) Once the above issues have been fully addressed, the clinical aspects of the topic should be greatly emphasized, as the article in its current state is balanced towards pathogenesis. Most of the clinical information is concentrated in the paragraph “Prevention and treatment of OSCC-associated thrombosis”. I suggest the following:
First, support your statements with more references. A review – even a non-systematic, narrative review – should collect available information from the literature. This is not an expert opinion or a Letter to the editor. Several statements lack references. Some instances, “DOAC are increasingly being used after several landmark trials showed them to be effective and safe therapeutical options for VTE.”, “n patients with OSCC surgical candidates CRS is another score that clinicians should consider” (also see my suggestion on re-writing this sentence); The recommended treatment after VTE diagnosis should continue for three to six months, always evaluating the bleeding risk. However, vitamin K antagonists (VKAs) and DOAC can be a safe alternative for long-term treatment if necessary. The same treatment of VTE would also be indicated for neck vein thrombosis.
Second, be more specific. Vague sentences that simply state “There is some information” but do not say, at least in short, what this information is, do not help the reader at all, despite the stated aim of this review to provide an overview on the topic: “Some good clinical practices for managing VTE in surgical oral cancer patients are also available and are given by societies of head and neck surgeons or ear, nose, and throat specialists”. So, what are clinical practices? For instance, in what exactly do they differ from the overall management of VTE (in cancer or non-cancer patients) or from the management of VTE in OTHER types of cancer? Or do they not differ at all? In this case, there are references, but some information should be provided.
Third, expand. A couple of examples of points of interest. Over the whole article, there is no information on the distinction between the two types of VTE of major clinical interest, deep vein thrombosis and pulmonary embolism. Is something known on whether oral cancer is associated with one more than the other? The prognosis of PE is worse; therefore, this would be clinically relevant. There is no information on the prognosis of patients with oral cancer-associated VTE. In addition to the instruments to assess the risk of VTE in cancer, is something known on the prognosis of patients with oral cancer-associated VTE? Does it differ at all from that of VTE associated with other types of cancer? Are there suggestions for follow-up strategies in the literature or in guidelines?
Fourth: if something is now known because it has not been investigated yet, then state it explicitly. It also belongs to the roles of a review to detail what should be studied next and help to set priorities for future research.
OTHER ISSUES
> Figure 1 is simply a table in figure version. Displaying a table as a figure has no advantages; it simply decreases readability, as text in a figure has a lower resolution than a table. Please transform it into a table using Microsoft Word.
> Abstract: “Cancer-associated thrombosis is a condition in which relevance has been increasingly recognised both for physicians that deal with venous thromboembolism and for oncologists.” May be simplified and shortened by simply avoiding the relative clause, and improved with some word changes, into “The relevance of cancer-associated thrombosis has been increasingly recognised both for physicians who treat patients with venous thromboembolism and for oncologists.”
> Abstract: “This relationship has been little investigated” may be unclear, as the previous sentence is long and the antecedent may not be obvious. I suggest rephrasing the previous sentence and this one into “Active cancer accounts for 20% of the overall incidence of venous thromboembolism. While oral squamous cell carcinoma is the most common form of oral cancer, its relationship with venous thromboembolism has been little investigated.”
> Abstract “Our aim is to spread its knowledge in order to improve patient care” can be improved into “In this review, we provide an overview of the topic with the aim of raising awareness on its relevance for patient care”.
Author Response
The Authors present a review on the association between oral squamous cell carcinoma (OSCC) (the most common type of oral cancer) and venous thromboembolism (VTE). The main argument for this publication holds: indeed, while much is known on the association of cancer with VTE in general and of specific types of cancer (for instance, solid vs. hematogenous cancers) with VTE in particular, little has been written on the specific topic of OSCC and VTE. An overview of what little information is available would be clinically useful. In addition, the Authors present some interesting information on the biological characteristics of SCC and their potential association with thrombosis that may have implications for prevention and for future research. However, the article in its present form is not suitable for publication and should be revised before it can be considered for Cancers. The main problems are the organization, formatting, and presentation of the information. While the English is in general adequate, the article should be proof read because of some redundancies and style issues. In addition, the clinical relevance of the topic and of available evidence should be emphasized much more strongly. I present the major issue in detail below.
MAJOR ISSUES
- While the written English is generally adequate, with no major grammar issues, the style should be considerably improved. In particular, several paragraphs and sentences display redundancy: removing unnecessary sentences or phrases would greatly increase readability, as required by scientific style. I picked a couple of examples, but these examples are non-exhaustive and addressing the examples only would not solve the problem… Therefore, before anything else, I would ask the Authors to reduce by 25% the total length of the manuscript by carefully going through it and removing unnecessary words, phrases, and sentences… Clarity is further affected by at times inadequate style and formatting. Again, some instances that are non-exhaustive… The paragraph beginning with “Notwithstanding” is not entirely clear. Please check carefully and rephrase the whole period as clearly as possible…The use of new lines is questionable. Each paragraph should discuss a topic or a previous study, and a new line should signal that the discussion moves to a new topic or a new study…There is a proliferation of “either” in the sentence . Please correct…There are several typos: …“Actually...” Please remove… Word order is not always correct…It is not possible to write “In this paper” at a location in the sentence in which no paper has been cited (only a group of Authors). If anything, one could write “An important mechanism inducing coagulation and involving TF was described in a study by Adesanya et al., which showed that (...)”…Commas are not always used appropriate…Therefore, my second suggestion is to have a professional proofreader or a native English speaker go through the article to solve these problems. They can also help with step 1 above (the reduction of 25% of the length of the article in its present form).
- We would like to thank the reviewer for his critical comments, which enabled the paper to be improved. According to his suggestion, we have greatly modified the body of the text by avoiding redundancy, improving the style and reducing the content by about 20%. According to his precise instructions, we edited some paragraphs, corrected typing errors, improved syntax and grammar with the help of a native speaker expert and proofreader. Hopefully, the article will read more fluent and in accordance with the reviewer's expectations
- Once the above issues have been fully addressed, the clinical aspects of the topic should be greatly emphasized, as the article in its current state is balanced towards pathogenesis. Most of the clinical information is concentrated in the paragraph “Prevention and treatment of OSCC-associated thrombosis”. I suggest the following: First, support your statements with more references. A review – even a non-systematic, narrative review – should collect available information from the literature. This is not an expert opinion or a Letter to the editor. Several statements lack references. Some instances, “DOAC are increasingly being used after several landmark trials showed them to be effective and safe therapeutical options for VTE.”, “n patients with OSCC surgical candidates CRS is another score that clinicians should consider” (also see my suggestion on re-writing this sentence); The recommended treatment after VTE diagnosis should continue for three to six months, always evaluating the bleeding risk. However, vitamin K antagonists (VKAs) and DOAC can be a safe alternative for long-term treatment if necessary. The same treatment of VTE would also be indicated for neck vein thrombosis. Second, be more specific. Vague sentences that simply state “There is some information” but do not say, at least in short, what this information is, do not help the reader at all, despite the stated aim of this review to provide an overview on the topic: “Some good clinical practices for managing VTE in surgical oral cancer patients are also available and are given by societies of head and neck surgeons or ear, nose, and throat specialists”. So, what are clinical practices? For instance, in what exactly do they differ from the overall management of VTE (in cancer or non-cancer patients) or from the management of VTE in OTHER types of cancer? Or do they not differ at all? In this case, there are references, but some information should be provided. Third, expand. A couple of examples of points of interest. Over the whole article, there is no information on the distinction between the two types of VTE of major clinical interest, deep vein thrombosis and pulmonary embolism. Is something known on whether oral cancer is associated with one more than the other? The prognosis of PE is worse; therefore, this would be clinically relevant. There is no information on the prognosis of patients with oral cancer-associated VTE. In addition to the instruments to assess the risk of VTE in cancer, is something known on the prognosis of patients with oral cancer-associated VTE? Does it differ at all from that of VTE associated with other types of cancer? Are there suggestions for follow-up strategies in the literature or in guidelines? Fourth: if something is now known because it has not been investigated yet, then state it explicitly. It also belongs to the roles of a review to detail what should be studied next and help to set priorities for future research.
A: We have carefully examined the reviewer's requests and made several changes. Unfortunately, we did not find much scientific literature on this topic, so we have tried to better select the literature referring to the OSCC-thrombosis association. We balanced the content of this article and added information on the problem of pulmonary embolism (unfortunately there is not much!). We have fully embraced the fourth point: making explicit the “grey areas” that need future studies. Finally, we modified the abstract according to suggestions.
Q: Figure 1 is simply a table in figure version. Displaying a table as a figure has no advantages; it simply decreases readability, as text in a figure has a lower resolution than a table. Please transform it into a table using Microsoft Word.
A: Ok, done. We have removed the figure. Furthermore, after major changes to that paragraph, we did not consider a table to be necessary

Round 2
Reviewer 1 Report
The authors appropriately revised the manuscript, which seems much improved.
Author Response
Thank you very much for improving our paper with your comments
Reviewer 2 Report
The Authors have carefully addressed the points raised in the first round of review and the article has improved accordingly. I particularly appreciate the reduction in size and the removal of redundant information as well as the considerable improvements in the management section.
I noted a number of sentences that still merit attention and should be addressed.
Abstract: “Moreover, the limited available information causes a low attention (…)”: technically, this is an opinion and not a fact, as it is not supported by evidence. Please hedge into “Moreover, the limited available information may result in poor attention to/awareness of (…)”.
“inflammatory response typical of OSCC may trigger thrombosis” (remove “be” and “for”).
“Not by chance”: remove, for several reasons: 1) Not typical English (more typical English would be: “It is no coincidence that…”), 2) reflects opinion, rather than fact (association is not causation!), 3) superfluous (the point remains).
“an increasingin d-dimerblood valuemay suggestfurther investigations to exclude 154thrombosis.” Correct into: “Increased blood D-Dimer may prompt investigations to exclude thrombosis”.
“we confirm the uselfulness of examining neck vessels by echo color Doppler with aim of excluding jugular vein thrombosis mostly[6]”: as already discussed in the first round of review, a scientific review should list known facts and hypotheses, not the Authors’ opinion. In the scientific world, the place for Authors’ opinions are Letters to the Editor and Editorials. Please re-write into “Doppler echocardiography may be used to examine neck vessels and exclude, in particular, jugular vein thrombosis [6]”.
“In a recent study, a potential mechanism that may explain this paradox was explained using an original mouse model of tongue squamous cell carcinoma[26].” Please remove “original”.
“platelet accumulation and fibrin generation in OSCC resulted highly inhibited”: replace “resulted” with “were”.
“We also hypothesized that the low incidence of thrombosis in OSCC was due to a difference with general risk factors for cancer-associated thrombosis”: why do the Authors suddenly switch to the first person? Are they referring to a study that they themselves conducted? Then it is legitimate to say “We”, but please state that more clearly in a sentence with a clear structure, such as “Our research group recently conducted a study investigating… We hypothesized that… we found that…” The Authors write that correctly in a sentence later on: “In our clinical experience…” is clear. If, instead, the sentence does not refer to a study conducted by the Authors of this review themselves, please correct the subjects (“The Authors of a recent study hypothesized..”, “Several studies in the recent literature”; or whatever applies, it is not clear in the current form).
“This venous clot is probably a “tumor” thrombus, not a typical clot”: I would re-write into “This venous mass may not be a fibrin clot, but rather a mass of tumour cells”.
“Moreover, in our experience a thrombosis of external and/or internal jugular vein can be cause of surgical unsuccessful, especially after microvascular reconstruction.” Please refrain from making causality statements based on such tenuous evidence. Good reporting of epidemiological and biostatistical findings required statements of causality to be avoided even in prospective longitudinal studies, let alone in unpublished clinical series without even a statistical testing. Association is not causation! Please be careful and rewrite: “Moreover, in our experience surgical failure was more common in patients with a thrombosis of external and/or internal jugular vein, especially after microvascular reconstruction”.
“main predictors for VTE were elderly, prior thrombosis, obesity and bloodtransfusion.”: “elderly” is an adjective, not a noun. Please write “Older age”.
“with a high (26%) but not significant number of vascular events [31]”. What do the Authors mean with “not significant number of vascular events”? “Significant” should be reserved for statistical association. Please rewrite and state more clearly what you mean.
“However, in this study it was very interesting to have adopted a risk stratification score for VTE known as Caprini risk score”. This is quite long-winded and not entirely grammatically incorrect (why the infinitive clause “to have adopted”?). Why not simply write “Of note, this study tested the association of the Caprini score with VTE”.
“the literature search on the OSCC-pulmonary embolism association is poor in information”: maybe more clear as “The literature search on the association between OSCC and pulmonary embolism did not return results”.
“Its management will be discussed in the last paragraph”: Remove.
“Among risk factors,”: remove.
“Cancer patients have an estimated almost 7-fold increased risk of developing chemotherapy associated-thrombosis”: compared with whom? Non-cancer patients? But that is obvious, is it not? If cancer patients have an estimated almost 7-fold increased risk of developing chemotherapy associated-thrombosis compared with non-cancer patients, this is not surprising: non-cancer patients do not receive chemotherapy at all. This is similar to writing that women have an X-fold higher chance than men than developing uterus carcinoma: the risk ratio must be positive (and tend to infinity), as men do not even have a uterus (except for some rare chromosomal aberrances). I am therefore not sure that that is the point of reference (32): please check and rewrite.
“Indeed, thrombosis can be a complication in oral cancer surgery and often leads to flap failure.” See my previous comment on causality: can the Authors prove that thrombosis is the cause of flap failure, or should they not rather simply write that “is associated with flap failure”?
Author Response
We thank the reviewer for his precious suggestions. We have agreed to all his points aimed at improving the clarity of the paper and made all the requested changes.
